# Pareto Optimal Learning from Preferences with Hidden Context

## Abstract

Ensuring AI models align with human values is essential for their safety and functionality. Reinforcement learning from human feedback (RLHF) leverages human preferences to achieve this alignment. However, when preferences are sourced from diverse populations, point estimates of reward can result in suboptimal performance or be unfair to specific groups. We propose Pareto Optimal Preference Learning (POPL), which enables pluralistic alignment by framing discrepant group preferences as objectives with potential trade-offs, aiming for policies that are Pareto-optimal on the preference dataset. POPL utilizes lexicase selection, an iterative process that selects diverse and Pareto-optimal solutions. Our theoretical and empirical evaluations demonstrate that POPL surpasses baseline methods in learning sets of reward functions and policies, effectively catering to distinct groups without access to group numbers or membership labels. We verify the performance of POPL on a stateless preference learning setting, a Minigrid RL domain, Metaworld robotics benchmarks, as well as large language model (LLM) fine-tuning. We illustrate that POPL can also serve as a foundation for techniques optimizing specific notions of group fairness, ensuring safe and equitable AI model alignment.

## 1 Introduction

For both safety and functionality, it is critical for AI models to align with the values of human users and stakeholders. Recently, reinforcement learning from human feedback (RLHF) (Christiano et al., 2017) has emerged as an effective mechanism for model alignment, using preferences to capture human values. However, when preferences are sourced from large groups of potentially diverse people, methods that rely on point estimates of human values are bound to either be suboptimal for all groups or unfair to certain groups, both of which are problematic in their own ways.

In this work, we build upon the notion of hidden context proposed by Siththaranjan et al. (2023) and focus on the problem of Reinforcement Learning from Human Feedback with Hidden Context (RLHF-HC). Hidden context refers to information that is unavailable to a preference learning system yet affects the preferences given. For example, a person's dominant hand might determine on which side they would prefer a robotic assistant to hand them an object. Under this formulation, our goal is to build a *set* of policies that contains the optimal policy under the reward function for each group of people. In practice, we see two clear use cases of such a set of policies. First, they can be selected from at test time to find an optimal policy for a given user without in a few-shot manner. Second, this set can be used to measure and ensure fairness between groups. Minimizing risk with respect to this diverse distribution of policies ensures that no specific group is disregarded in the risk measurement–thus enhancing safety.

Preferences with hidden context may be contradictory *i.e.*, not mutually satisfiable by a well-regularized policy or reward function. So, we propose to frame these preferences as objectives with potential trade-offs between each other. With this re-framing, the optimal policy for each individual hidden context group would be Pareto-optimal (non-dominated) on the dataset of preferences. With this in mind, we propose Pareto Optimal Preference Learning (POPL), where we learn a set of reward functions or policies (directly) that are optimized towards being Pareto-optimal with respect to the set of preferences given by a potentially diverse set of human annotators. To do this, we use an iterative selection process known as lexicase selection (Spector, 2012), which has been shown

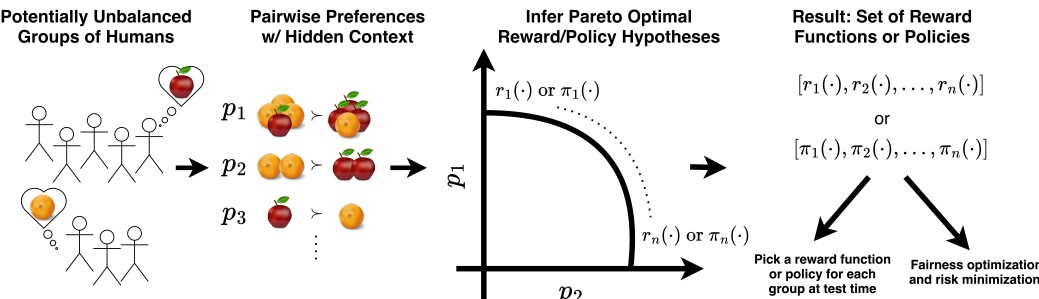

Figure 1: An outline of the proposed Pareto Optimal Preference Learning (POPL) framework. Given a set of pairwise preferences over trajectory segments from groups with potentially different ground truth reward functions, we infer a set of reward functions or policies that captures each group's ground truth, without group membership labels. To do this, we frame reward inference as multi-objective optimization, where each preference forms a single objective, and find a set of Pareto-optimal reward functions or policies.

under mild assumptions to select individuals that are both Pareto-optimal and diverse. An outline of our method can be found in Figure 1. Our contributions can be summarized as follows:

- We extend the problem of Reinforcement Learning from Human Feedback with Hidden Context (RLHF-HC) introduced by Siththaranjan et al. (2023), addressing critical limitations in preference learning for sequential, time-based domains, as opposed to contextual bandits.

- We derive theoretical results proving that optimal reward functions and policies for hidden context groups are inherently Pareto-Optimal with respect to the given preferences, establishing a rigorous mathematical basis for our approach.

- We develop "Pareto-Optimal Preference Learning" (POPL), a framework leveraging lexicase selection to generate a set of Pareto-Optimal reward functions or policies. POPL ensures diverse, group-specific alignment with human preferences, enabling robust personalization and fairness.

- We demonstrated POPL's superiority over strong baselines in diverse settings, including:
  - Minigrid RL: Policy learning in grid-based decision making dsomains.
  - Metaworld: Balancing safety and speed in 3D robotics manipulation tasks.
  - LLM Jailbreaking Detection: Mitigating harmful outputs by aligning preferences for both helpfulness and harmlessness (without labels).

- We showcased POPL's ability to efficiently scale to high-dimensional tasks, such as those involving LLMs, while maintaining computational efficiency. POPL achieves robust results with pre-trained models, making it broadly applicable across domains requiring fairness, alignment and diversity.

## 2 RELATED WORK

**Diversity in Human Preferences**    Data used for RLHF systems often comes from multiple people, who are diverse in their preferences and values (Bobu et al., 2023; Peng et al., 2023; Biyik & Sadigh, 2018; Santurkar et al., 2023). This data, when considered in its aggregated form, can not be captured perfectly by a decision-making model that relies on a point estimate of utility (Casper et al., 2023). These models try to find a single reward function that is most likely, which is often not the optimal reward function for any one single person. When the groups are not perfectly balanced, the minority groups might be underrepresented in the inferred reward function (Siththaranjan et al., 2023; Feffer et al., 2023; Kirk et al., 2023; Myers et al., 2021) or simply treated as noise (Baumler et al., 2023). There have been attempts at explicitly modeling different people with different levels of expertise (Gordon et al., 2021; Daniels-Koch & Freedman, 2022; Gordon et al., 2022; Barnett et al., 2023), but these methods generally rely on concrete ways to distinguish between groups.

**Dealing with this Diversity** In the context of RL, Myers et al. (2021) outlines an approach to learning a multi-modal reward function from online interaction between a human expert and a preference learning system. Ramé et al. (2024) learn a set of reward models by optimizing for diversity amongst the outputs. While similar to our approach, we also aim to align our reward models with hidden context groups through optimizing for Pareto-optimality. For generative AI models and LLMs, there have been a variety of studies attempting to align large models with diverse human preferences. Chakraborty et al. (2024) and Siththaranjan et al. (2023) learn a mixture of preference distributions or a parameterized reward distribution, respectively. However, both these techniques operate under a contextual bandit setting which results in sub-optimal performance when used in the more general RL setting (discussed further in Section 3). Bradley et al. (2024) and Ding et al. (2024) leverage fine-tuning to improve the diversity of model responses for better alignment and creativity, which do not directly address the ambiguity and hidden context in human preferences. Jang et al. (2023) and Dai et al. (2023) elicit preferences specifically along different dimensions in order to cater custom reward functions for users in test time, and to be safe with respect to conflicting objectives, respectively. Whilst we also aim to cater reward functions in test time as well as optimize fairness between groups, we do not have access to labels regarding the context of the preferences generated. Finally, Rame et al. (2024) also generates a set of Pareto-optimal reward functions. However, in their setting, the system has access to ground truth reward functions for each group, and the Pareto-front is generated through weight interpolation between these functions.

**Bayesian Reward Inference** Bayesian Reward Extrapolation (B-REx) (Brown et al., 2020b) instead learns a distribution of reward models from pairwise human preferences. B-REx is then able to perform Bayesian inference using MCMC (MacKay, 1992) to sample from the posterior of reward functions. With this distribution, a practitioner can establish high confidence performance bounds that can be used to assess risk in evaluated policies as well as detect reward hacking behaviors. However, B-REx and other reward inference methods often rely on a faulty assumption that humans provide preferences in a Boltzmann-rational way.

## 3 PRELIMINARIES

**Learning from Human Preferences** Reinforcement learning from human feedback considers human preferences over trajectories (or more generally, outputs of a model) in order to learn a reward model or policy that respects the preferences (Brown & Niekum, 2019; Rafailov et al., 2024; Hejna et al., 2024; Casper et al., 2023; Finn et al., 2016).

In order to learn meaningfully from human preferences, one must characterize how preferences are generated from some parameterized preferences model $P(\sigma_i \prec \sigma_j)$. Usually, this preference model is based on the notion of Boltzmann-rationality, where humans generate preferences in accordance to the Bradley-Terry (BT) model (Bradley & Terry, 1952). The probability of pairwise preference $(\sigma_i \succ \sigma_j)$ between two trajectories segments given some utility function $f(\sigma)$ can be written as

$$P(\sigma_i \succ \sigma_j) = \frac{e^{\beta f(\sigma_i)}}{e^{\beta f(\sigma_j)} + e^{\beta f(\sigma_i)}} \tag{1}$$

where $\beta$ models the confidence in the preference labels. $\beta \to \infty$ signals that the preference provider is perfectly rational, and $\beta = 0$ signals that preferences are random. The BT model is used in many fields, such as psychology (Baker et al., 2009; Goodman et al., 2011; Goodman & Stuhlmüller, 2013). However, this model does not perfectly capture the mechanisms driving these preferences (Ghosal et al., 2023; Jeon et al., 2020; Knox et al., 2022; Bobu et al., 2020; Lee et al., 2021).

**Hidden Context** Siththaranjan et al. (2023) introduce the problem of preference learning with hidden context. This is the idea that preferences are generated not only based on the exponential utility (partial return or regret), but also on some latent hidden context variable $z$. This variable is not accessible to preference learning systems and poses a challenge as it is often the case that this variable results in breaking the assumption that preferences are generated Boltzmann-rationally.

**Marginalized Distributional Preference Learning** In order to account for hidden context in the preferences learned, Siththaranjan et al. (2023) introduce Distributional Preference Learning, which

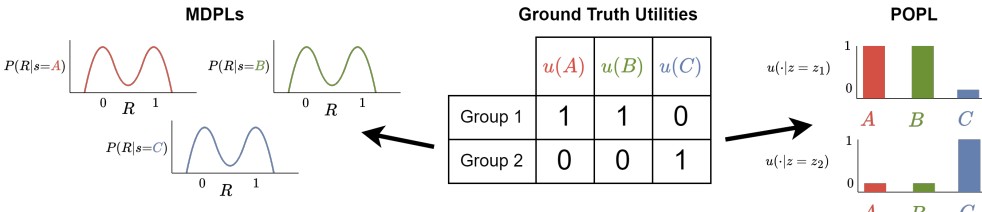

Figure 2: An example of a situation where using POPL is preferable to using a Marginalized Distributional Preference Learning (MDPL) system. Due to the fact that these systems marginalize over the hidden context $z$ for each state, MDPLs are unable to be sensitive to persistent annotator identity. MDPLs represent the distribution of utility values in a column-wise fashion, or maintain a distribution of utilities for each state, that is decoupled from that for other states. Therefore, the utility for both groups of the trajectory $AB$ is indistinguishable from that for $BC$ by an MDPL. POPL, on the other hand, represents the distribution row-wise, finding a set of utility functions that should include the ground truth for each group. In this case, POPL can represent the fact that $AB$ is an unfair trajectory and $BC$ is fair, whereas MDPLs are unable to make this distinction.

relies on a single model of utility $u(s|z)$ to output a distribution of utility assignments $u(s)$ for each state $s \in \mathcal{S}$, marginalizing over the hidden context variable $z$. In other words, they are able to represent the marginalized probability $P(R|s)$ of a specific utility $R$ in a state $s$. Herein, we will refer to a model that outputs a distribution per state as a Marginalized Distributional Preference Learning (MDPL) system.

Due to the marginalization process inherent in these systems, the utility function is unable account for *persistent annotator identity*—the fact that hidden context transcends a single preference annotation. In a contextual bandit setting such as those often found for finetuning LLMs (Rafailov et al., 2024), this is not an issue, as determining that an output has high risk simply depends on the distribution of rewards attributed to that specific state. However, in sequential tasks, where there are relationships between preferences at different times, it is important to maintain full, coherent, reward functions or policies for each group.

An example of how using an MDPL can lead to fairness issues is outlined in Figure 2. There are two groups that have different utilities for three states A, B and C. MDPLs fail to differentiate between trajectories like AB, BC, and AC, which have distinct fairness profiles and utilities for different groups, as their representation marginalizes over group-specific utilities.

**Contrastive Preference Learning** Contrastive Preference Learning (CPL) (Hejna et al., 2024) learns a policy directly from preferences without needing to learn an intermediate reward function. This method uses a regret-based model of preferences rather than the standard partial return interpretation. The probability of a preference under a candidate policy can be written as the ratio of the exponentiated sum of log-likelihoods of the chosen segment to the disregarded segment. We choose CPL over Direct Preference Optimization (DPO) Rafailov et al. (2024) as DPO can be derived as a special case of CPL with trajectories of length 1, starting from the same state. Furthermore, POPL is designed to be used in a variety of sequential, time-based domanis, but DPO and other contemporary RLHF methods restrict themselves to contextual bandit settings (such as in large language models). CPL, on the other hand, overcomes these limitations Hejna et al. (2024).

## 4 PROBLEM STATEMENT AND THEORETICAL FOUNDATION

We operate in a common RLHF setting in which, given a dataset $D = \{\sigma_1, \cdots, \sigma_m\}$ of trajectory segments and a set $\mathcal{P} = \{(i, j) : \sigma_i \succ \sigma_j\}$ of pairwise preferences over these segments, we wish to infer an unknown reward function $r : \mathcal{S} \mapsto \mathbb{R}$ that respects the preferences. This reward function represents an assignment of utility $r(s)$ to each state $s$ in the state space $\mathcal{S}$. $r$ can then be used as the reward function to train a policy $\pi(a|s) : \mathcal{S} \times \mathcal{A} \mapsto [0, 1]$ with RL.

In light of our discussion in section 3, we re-frame the problem of preference learning with hidden context as follows. The goal is to learn a set $\Pi = \{\pi_1, \pi_2 \cdots, \pi_n\}$ of policies such that, for the hidden context group represented by a variable $z \in \mathcal{Z}$, there is a policy $\pi_z \in \Pi$ that is the optimal policy for the ground truth reward function $r_z$ for the group. Note that this can be accomplished by standard (reward-based) RLHF (experiments in sections 6.1 and 6.4) or direct (reward-free) RLHF (experiments in sections 6.2 and 6.3). For the standard approach, a series of reward functions $R = \{r_1, r_2 \cdots, r_n\}$ are first learned from preferences, then used to train $n$ optimal policies. In the direct approach, the policies are learned directly from preferences such as done by Hejna et al. (2024) and Rafailov et al. (2024).

We now show that optimal policies for hidden context groups are Pareto-optimal with respect to the set of preferences given by all annotators. Therefore, recovering the set of pareto-optimal policies is a viable way to solve the RLHF-HC problem formatted above.

**Definition 1 (Policy passing preference).** A policy $\pi(a|s) : \mathcal{S} \times \mathcal{A} \mapsto [0, 1]$ passes a preference $(\sigma_i \prec \sigma_j)$ if the probability of the preferred segment $\prod_{(s,a) \in \sigma_j} \pi(a|s)$ is higher than the probability of the other segment $\prod_{(s,a) \in \sigma_i} \pi(a|s)$. Or, equivalently, if $\sum_{(s,a) \in \sigma_j} \log \pi(s,a) > \sum_{(s,a) \in \sigma_i} \log \pi(s,a)$.

**Definition 2 (Policy-set-relative Pareto-optimality).** A policy $\pi(a|s) : \mathcal{S} \times \mathcal{A} \mapsto [0, 1]$ is Pareto optimal with respect to a set of preferences $\mathcal{P}$ relative to a set of other policies $\Pi = \{\pi_1, \pi_2, \cdots, \pi_n\}$ if and only if there exists a preference $(\sigma_i \succ \sigma_j) \in \mathcal{P}$ that only $\pi$ passes out of all policies in $\Pi$

**Definition 3 (Hidden context group).** A hidden context group is a group of $m$ annotators, each with their own reward function $r_1, r_2, \ldots, r_m$ that identically rank the segments $\sigma \in \mathcal{D}$.

**Definition 4 (Optimal policy for hidden context group).** An optimal policy $\pi_g^*$ for a hidden context group $g$ is an optimal policy in a given environment (MDP/R) using the group's implicit ground truth reward function $r_g$ as the reward function.

**Definition 5 (Contradictory preferences).** A pair of preferences $(\sigma_i \succ \sigma_j)$, $(\sigma_x \succ \sigma_y)$ are contradictory under a specific policy regularization scheme if the likelihood of any policy that satisfies both is lower than the likelihood of a policy that satisfies either.

With no regularization, a policy that satisfies two preferences should have a higher likelihood than one that satisfies one but not the other. This is because the likelihood of a certain policy is related to the total number of preferences passed by it. Passing two preferences would therefore elicit a higher likelihood than passing just one of them. However, if by satisfying both preferences, the models would need to incur a much greater regularization loss, it is likely that these two preferences came from individuals with differing hidden contexts.

**Theorem 1.** In a completely noiseless setting, all policies that are optimal for specific HC groups are Pareto optimal with respect to the set of all preferences $\mathcal{P}$ generated from all the groups, and the space of all possible policies $\Pi = \{\pi | \pi \in \mathcal{S} \times \mathcal{A} \mapsto [0, 1]\}$

A proof of Theorem 1 can be found in the appendix. In essence, a set of Pareto-optimal policies must each satisfy a unique set of mutually satisfiable preferences (ones that do not contain a contradictory preference). As such, the optimal policies for a group with hidden context would also be Pareto-optimal.

## 5 PARETO OPTIMAL PREFERENCE LEARNING

In this section, we outline an algorithm that can be used to generate a set of polices or reward functions that align with the preferences of different groups of people. We introduce lexicase selection, a method that can select candidate hypotheses that lie on the Pareto front. Our population represents a belief distribution that is updated based on observed evidence. Lexicase selection continually narrows the hypothesis space based on selection criteria, effectively 'learning' which policies or reward functions hold promise given the current (hidden-context-laden) data.

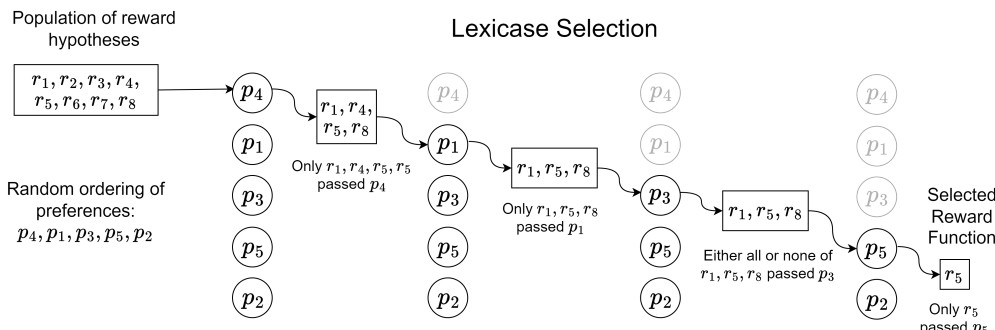

Figure 3: Lexicase selection being used to select a single candidate hypothesis. Starting with a random ordering, the pool of reward hypotheses is filtered down based on the preferences in order, until a single individual remains or we run out of preferences. The resulting reward function is added to the next pool, and this process is repeated (with new shuffles) to fill the population.

**Lexicase Selection for Pluralistic Outcomes**     To obtain a set of Pareto optimal policies, we adopt the idea of lexicase selection (Spector, 2012; Helmuth et al., 2015), which uses a random ordering of metrics for each selection event, with only the candidates that perform the best on each successive metric retained for filtering by the remaining metrics. This process is repeated until all metrics are exhausted or a single individual remains. Through this process lexicase selection prioritizes, over multiple selection events, each particular metric and each particular combination of metrics to the exclusion of all others. We can consider this to be a "particularity" approach for achieving pluralistic outcomes (Spector et al., 2024).

In the setting of preference learning, each metric corresponds to a preference sourced from a human with hidden context. The preference is 'passed' if the candidate policy correctly ranks the pair of segments in the preference corresponding to that metric (formally defined in Section 4). If no individuals in the current pool pass the preference, all individuals make it through this selection step. With this feature, contradictory preferences are addressed by giving priority to the first preference in the shuffle. Each random shuffle of preferences therefore results in a diverse profile of reward functions being selected for. Figure 3 shows an example of a single selection event.

A key property of lexicase selection is that it selects candidates that are Pareto-optimal relative to a starting set of candidates (as opposed to *all possible candidates*). These individuals tend to spread the *corners* of the Pareto front and thus be diverse (La Cava et al., 2019). Lexicase selection also gives individuals that are good at more subsets of things greater weight. This idea has been utilized in many machine learning optimization problems for improving generalization, as shown in recent work (Ding et al., 2022; 2023; Ni et al., 2024; Boldi et al., 2023; Ding & Spector, 2022).

**Overview**     With a method to select Pareto-optimal candidates such as lexicase selection in hand, one can infer a set of reward functions or policies directly from preferences. Initially, a random set of candidate models is created. Then, the chosen method is applied to select (with replacement) the Pareto-optimal candidates from this random starting set. This pool is perturbed by adding random Gaussian noise, generating a new set of candidates. The selection and perturbation steps are repeated iteratively until the average performance converges, or a fixed number of iterations is passed. where a distribution of hypotheses are The final set of candidates should align with the preferences of hidden context groups. A full overview of our algorithm can be found in Appendix A.

## 6 EXPERIMENTS

In this section, we detail our experimental results to validate the proposed POPL method. To verify that POPL can work in a large variety of settings at different scales, as well as for generating both reward functions and policies, we perform four sets of experiments. A synthetic, stateless experiment (reward inference), a Minigrid RL environment (policy inference), a Metaworld robotics environ-

ment (policy inference) and LLM finetuning from human preferences (reward inference). Further implementation details are provided in Appendix C.

**Baselines** Throughout our experiments, we will use 3 main baselines. In the experiments on reward function inference, we use Bayesian Reward Extrapolation (B-REx) (Brown et al., 2020b) as a baseline, as it generates a large set of reward function hypotheses (i.e., candidate models) based on a Boltzmann-rational likelihood function, and has demonstrated efficacy in RL domains. For our policy inference experiments, we compare to Contrastive Preference Learning (Hejna et al., 2024) as it is a leading RLHF algorithm for sequential tasks. We also use a naive method of learning a *set* of policies based on CPL that we call Multi-CPL. In this approach, after pretraining, we fine-tune the last layer using the CPL objective multiple times to generate a large set of policies. Although we could do full network fine-tuning, we wanted to hold constant the trainable parameters available to each approach to ensure a fair comparison to POPL, which uses last-layer fine-tuning. Policy learning settings in this work model human preferences as being generated based on regret, as opposed to partial return (Knox et al., 2022). Including the policy inference experiments allows us to ensure our method is not sensitive to assumptions regarding how the preferences are generated. For our language model (contextual bandit) experiments, we compare to both B-REx and Distributional Preference Learning (DPL) (Siththaranjan et al., 2023), as well as standard the standard RLHF paradigm (Christiano et al., 2017), as these present a variety of approaches for generating reward models that can be used to ensure fairness across groups.

**Metrics** Given a set of reward functions or policies, we can verify how well they perform on the two downstream tasks we have identified for this work: personalization and fairness. For personalization, we inspect the content of the personalized policies or reward functions to verify their alignment with each hidden context group's preferences. For fairness, we ensure that no single group is having its values undermined (by taking low-probability actions) in an attempt to satisfy a different group. Although this is a relatively simple notion of fairness, this method could be extended to be compatible with other fairness optimization approaches Mehrabi et al. (2021).

### 6.1 SYNTHETIC STATELESS EXPERIMENT

The first set of experiments we perform will test whether POPL is able to recover a set of reward functions from a series of preferences generated with hidden context in a very simple stateless domain. Doing this, we are testing whether the fact that the outputs of lexicase selection are an approximation of the global Pareto-front significantly degrades the quality of reward functions generated. Then, we will select a personalized reward function for each group and compare them to the ground truth reward functions used to generate the preferences.

Following the synthetic experiments outlined by Siththaranjan et al. (2023), we compare B-REx and POPL on learning from preferences where with hidden context variable $z \sim \mathcal{B}(0.5)$ where $\mathcal{B}(0.5)$ is a Bernoulli distribution. The utility in this scenario can be modeled as

$$u(a, z) = \begin{cases} a & \text{if } a < 0.8 \\ 2az & \text{otherwise} \end{cases} \quad (2)$$

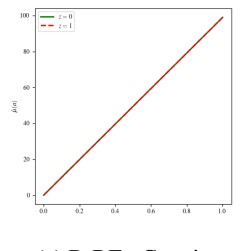
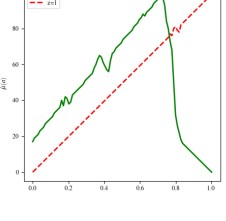

(a) B-REx Catering      (b) POPL Catering

Figure 4: (a) and (b) show the catered reward functions for each of the two hidden context groups $z = 0$, $z = 1$. From a set of reward functions that is inferred from a diversity of human preferences, we select a single reward function for each unique group with a small number of preferences (2% the size of the training set). POPL is able to cater for both groups, while B-REx is only able to cater for one of the two groups ($z = 1$, red line). For B-REx, the $z = 0$ (green) group's catered reward function doesn't capture the fact that any state $a < 0.8$ is preferred to any state $a \geq 0.8$.

In order to test whether POPL covers the hidden context groups, we inspect some selected reward functions for each group. We use a smaller set of the preferences that all have a shared hidden context, and select a reward function for each group. Figure 4 shows the results of catering a reward function for each of the hidden context classes $z = 0$ and $z = 1$. Due to B-REx using the Boltzmann rationality assumption, it concentrates much of the distribution on the $z = 1$ case, and does not

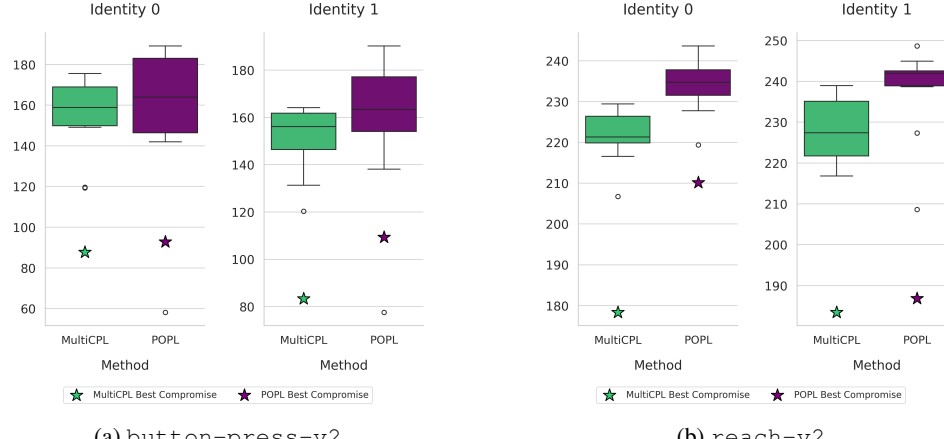

(a) `button-press-v2`       (b) `reach-v2`

Figure 6: Metaworld Policy Inference Results. Box plots outline the performance of the best (catered) individual from each population on both identities across 10 random seeds. We also show the average performance of the best "compromise," or the single policy that does the best across both identities. POPL tends to have a higher catered policy performance across both identities, and also discovers a more fair compromise between values of the two groups.

capture the preferences given by the $z = 0$ group. POPL, on the other hand, is able to recover the reward functions for both groups from the learned distributions.

## 6.2 MINIGRID POLICY INFERENCE

After demonstrating POPL's efficacy in reward inference from preferences with hidden context, we perform a second set of experiments to verify whether 1) POPL is able to generate *policies* directly from preferences, and 2) POPL is able to perform in a sequential RL domain, where annotators' hidden context is persistent (i.e. potentially affects more than one segment preference annotation). The domain used in these experiments is outlined in Figure 5a. The agent (red triangle) must make it to the solid green goal tile as fast as possible. The agent must choose one of the two doors (top or bottom) to use to reach the goal. The hidden context groups in this scenario delineate whether the annotator inherently prefers the bottom or top door to be used to get to the goal (Figure 5b). The preferences were labeled according to the regret preference model from members of both groups (extracted from the optimal policy for each group's ground truth reward model).

After running this optimization, the state occupancy distribution for catered policies for each group can be found in Figures 5c and 5d for POPL, and 5e and 5f for MultiCPL. We find that POPL is able to successfully cater policies for both groups of people (as exhibited by policies reaching the goal via both doors), despite not having labels regarding their group membership. MultiCPL, on the other hand, is unable to cater a policy for Group 1.

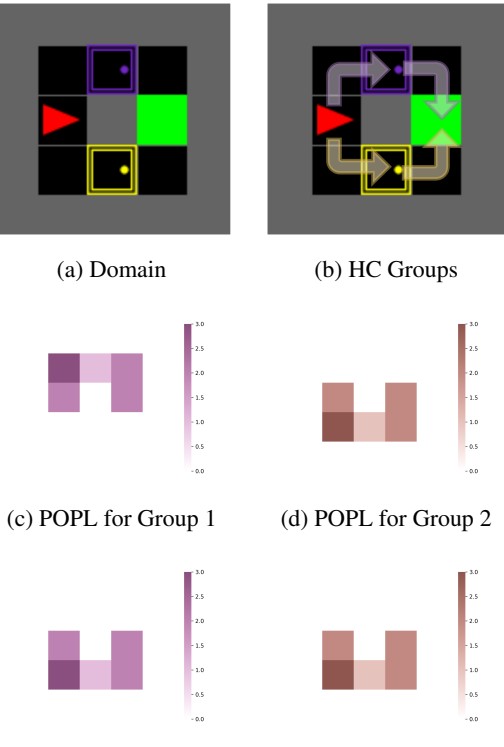

(a) Domain      (b) HC Groups

(c) POPL for Group 1  (d) POPL for Group 2

(e) MultiCPL for Group 1 (f) MultiCPL for Group 2

Figure 5: Minigrid experiments. Plots in (c), (d), (e) and (f) show average state occupancy for policies catered for each hidden context group. POPL is able to cater distinct policies for each group, while MultiCPL collpases to a single group's preferences.

## 6.3 METAWORLD POLICY INFERENCE

In order to verify how well POPL can infer policies in larger scale sequential environments, we include results performing policy inference on the Metaworld Robotics Benchmark (Yu et al., 2019). We artificially create two hidden context groups, one that prefers safe (low angular velocity) robotic movements, and one that prefers speed (having tasks completed quickly). We generate preferences from these two groups at random, and then compare POPL and MultiCPL's ability to cater individual policies for each group.

Figure 6 outlines the performance of all the policies generated by POPL and the MultiCPL baseline. We also include a case study comparing a single run of POPL and MultiCPL in Figure 7. POPL is able to generate policies that outperforms the MultiCPL and behavior cloning baselines. POPL finds policies that are

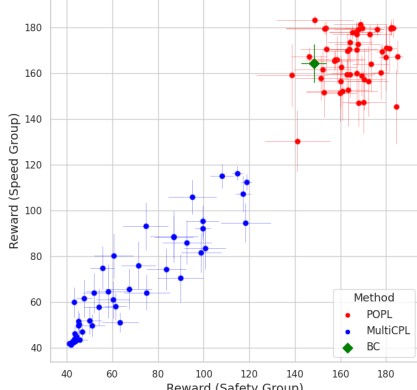

Figure 7: Case study for a single `button-press-v2` run. POPL finds policies that perform well under both ground truth reward functions. We also include a behavior cloning (BC) baseline, where the policies are simply trained to match the demonstrations.

maximally good for either group, as well as those that find strong compromises between the two group's values.

## 6.4 LANGUAGE MODEL EXPERIMENTS

In this section, we test the ability of POPL to scale to domains involving human annotations. We investigate whether POPL can be sensitive to hidden context in whether annotators prefer *harmless* or *helpful* responses (Bai et al., 2022). When reward models are trained on the entire set of preferences, whether they were generated based on helpfulness or harmlessness is hidden context, as this information affects preferences but is unavailable to a reward inference system.

Importantly, preferences based on helpfulness and harmlessness can often be contradictory. In fact, Wei et al. (2024) find examples of user prompts that directly pit these objectives against each other, leading a language model to output harmful outputs, a phenomenon known as *jailbreaking*. An RLHF system built with hidden context in mind would help detect jailbreaking before a harmful output would be given to a user. In the context of this work, a model that is susceptible to jailbreaking would be unfair to certain groups (compromising its efficacy for the harmlessness group in order to optimize for the helpfulness group).

Table 1 presents the jailbreak rates and helpfulness accuracy for standard RLHF, B-REx, DPL, and our proposed POPL. For B-REx and POPL, we generate a set of reward functions by extrapolating the last layer of a fine-tuned LLAMA-2-7b (Touvron et al., 2023) preference model. Default settings use the mean reward across the entire set. For fairness optimization, we use the 10th percentile of reward values across all the reward functions in the set.

The results indicate that B-REx's performance is inferior to standard RLHF, even when employing fairness-focused strategies using the lower quantile of rewards. This suggests that the likelihood estimated by the BT model does not adequately accommodate scenarios where preferences are in conflict, and B-REx fails to accurately approximate the distribution of rewards. POPL performs the best out of all methods without employing any fairness optimization. Given the high-dimensional nature of reward features in LLM tasks, a population-based approach is essential for accurately modeling and enhancing the diversity of reward hypotheses.

When compared to the current state-of-the-art, POPL outperforms Mean & Var DPL and competes closely with Categorical DPL. Notably, unlike DPL which requires training a new reward model with different outputs, POPL efficiently extrapolates directly from the last layer of pre-trained RLHF reward models, making it highly efficient and broadly applicable. For example, POPL can be applied to a pre-trained 7b-LLM reward model in under an hour on a single NVIDIA A100 GPU. Another advantage of POPL is its independence from assumptions about the distribution of reward hypothe-

Table 1: Results on LLM jailbreaks. POPL has the lowest jailbreak rate across all methods without any fairness optimization. For fairness optimization, POPL has a lower jailbreak rate than B-REx, standard RLHF, as well as Mean & var. DPL, and is competitive with categorical DPL.

| Method | Training data | Jailbreak rate (%) | Helpfulness acc. (%) |
|---|---|---|---|
| Standard | Helpful | 52.4 | 72.6 |
| Standard | Harmless | 3.7 | 49.5 |
| Standard | Combined | 25.1 | 68.2 |
| Mean & var. DPL | Combined | 30.5 | **68.4** |
| ↳ Fair | | 20.3 | **66.4** |
| Categorical DPL | Combined | 32.1 | 66.2 |
| ↳ Fair | | **13.4** | 66.2 |
| Bayesian REx | Combined | 28.3 | 67.5 |
| ↳ Fair | | 27.8 | 50.4 |
| POPL | Combined | **17.6** | 66.1 |
| ↳ Fair | | 15.0 | 65.7 |

ses. In contrast, DPL methods require a predefined reward distribution, such as the assumption of normally distributed rewards for Mean & Var DPL, or correctly sized bins for Categorical DPL.

## 7 CONCLUSION

When learning from human preferences for the sake of aligning to human values, systems often rely on point estimates of return or regret, limiting them to aligning to a single group of humans. Preferences, however, often come from distinct groups with diverse preferences. We have formalized this as the problem of preference learning with hidden context. Under this conception, a set of policies must be generated that contains the optimal policy for each distinct group.

To solve this problem, we relied on the concept of Pareto-optimality to generate a series of reward functions and/or policies that are optimal with respect to unique sub-sets of preferences. To optimize towards Pareto-optimality, we used a technique known as lexicase selection, that selects individuals from a large set based on a randomized (lexicographic) prioritization of the training data.

We verified that lexicase selection can be used to generate diverse distributions of either reward functions or policies that align with the diverse preferences that human annotators have. We evaluated and verified the performance of POPL in a variety of domains, including a synthetic stateless domain, a Minigrid RL domain, a Metaworld Robotics benchmark, and even language model jailbreak detection. Across these domains, we have demonstrated POPL's efficacy when compared to contemporary algorithms in dealing with hidden context in the preferences. Without modifications to the framework, POPL can be used to optimize for diverse reward functions or policies, and can work in stateless and sequential domains at a variety of scales.

One limitation of this work is the lack of use of gradients in training policies. The optimization procedure used after lexicase selection relies on random variations and repeated selections, which allows for effective trade-offs between exploration and exploitation of the preference landscape. Although empirically verified to work well, it may be possible to augment the core idea in future work to allow it to utilize gradients. Furthermore, a study into the conditions required for the output of the procedure to be globally Pareto-optimal could be instrumental.

### REPRODUCIBILITY STATEMENT

We are committed to the reproducibility of our results. We include full code to reproduce the results in this paper as supplemental material. This code includes dataset generation and the full POPL training pipeline. Furthermore, we outline experimental details needed to independently reproduce the results in Appendix C. The theory performed in Section 4 has proofs associated in Appendix B and assumptions outlined therein.

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

**Data:** A dataset of demonstrations $\mathcal{D}$ and a series of pairwise preferences $\mathcal{P}$
**Result:** A set of reward function or policy hypotheses

```
candidates ← randomly initialize p hypotheses
for iter 1 → N do
    for ind 1 → p do
        shuffled_prefs ← Shuffle(P)
        for pref in shuffled_prefs do
            old_subset ← candidates
            candidates ← subset of candidates that pass pref.
            // if all individuals have failed, we skip this preference
            // as it is likely to be contradictory with a previous preference
            if candidates contains no candidates then
                candidates ← old_subset
            end
            if candidates contains only one candidate then
                break
            end
        end
        candidate ← a random individual from candidates
        Append candidate to new population
    end
    candidates ← add random noise to candidates
end
return candidates
```

**Algorithm 1:** Pareto Optimal Preference Learning

## A  FULL ALGORITHM

Algorithm 1 gives an outline of a single step of Pareto Optimal Preference Learning (POPL).

## B  PROOFS

**Proof of Theorem 1 (Contradiction).**   Let us assume that there is a policy $\pi_z^*$ that is the optimal policy according to a hidden context group $z$. This means $\pi_z^*$ passes all the preferences compatible with the values of group $z$ and fails only the preferences that are not compatible with preferences given by group $z$. For sake of contradiction, we assert that this reward function is not Pareto optimal with respect to all other reward function candidates. This means there exists another reward function $\pi'$ that performs better than or equal to $\pi_z^*$ across every preference in $\mathcal{P}$, including those generated by group $z$. This is a clear contradiction as that would imply $\pi'$ is the optimal policy for group $z$ instead of $\pi_z^*$.

## C  IMPLEMENTATION DETAILS

In this section, we include more implementation details of our experiments.

### C.1  SYNTHETIC EXPERIMENTS

We follow the experimental procedure of Siththaranjan et al. (2023) in generating preferences, except modify their code such that we ensure that annotator identity is held constant for each preference. We use last layer finetuning on a neural network that is randomly initialized. We did not include any pre-training here to ensure that we are not pushing our reward models towards any modes before starting to train. We use a batch size of 2048 preferences, a step size of 0.1 and 10000 steps of MCMC for B-Rex. For POPL, we use a population size of 100 and a generation count of 100. We use a $\beta$ (confidence) value of 10, although have found that changing this value does not significantly affect B-REx's performance.

## C.2 MINIGRID EXPERIMENTS

For the Gridworld model experiments, we base our environment on the Minigrid package Chevalier-Boisvert et al. (2023). Demonstrations were generated by rolling out many checkpointed policies at different levels of performance, trained using Proximal Policy Optimization (PPO). Then, these demonstrations were annotated based on a high performing policy's action selection probabilities.

For MultiCPL and POPL, we use behavior cloning directly on the demonstrations for 1000 iterations with a batch size of 64 and a learning rate of 0.001 with the Adam optimizer Kingma & Ba (2014) as pretraining. The model architecture was a simple convolutional neural network that takes input from the agent's view window, and has a single fully connected layer with 128 nodes to output the 7 actions from the environment. For both MultiCPL and POPL, we use last layer fine-tuning. For MultiCPL, we use the CPL objective, a learning rate of 0.001, where each model in a population of 500 models is trained for 20 iterations. For POPL, we use a learning rate of 0.2, and 1000 total steps. We sample 640 preferences every 10 iterations (as we can cache the last layer features for this examples for improved performance), and sub-sample a batch of size 64 for each step of lexicase selection. For a fair comparison between these two approaches, we approximately hold constant total wall clock time on the same hardware. Given a final population of policies generated by POPL or MultiCPL, we select the top 10 models for each hidden context class as the catered policy for that group.

## C.3 METAWORLD EXPERIMENTS

For the Metaworld robotics benchmark (Yu et al., 2019), we create augmented reward functions with greater emphasis on speed or safety, respectively. For the speed reward function, we add a penalty of $\frac{10}{T}$, where $T$ is the maximum timesteps allowed for that environment, for every timestep until the goal (as defined by the metaworld environment) is met. For the safety reward function, we add a penalty of $10 \cdot \|\Omega\|_2$ where $\Omega$ is the angular velocity of all the robot's joints. We also include, for each group, the reward from the other group, weighted with 0.1 instead of 10.

We generated demonstrations by training optimal policies on each task studied using Proximal Policy Optimization (Schulman et al., 2017) with the Stable Baselines package (Raffin et al., 2021). Every 100,000 steps, we cached the policy parameters to be used to generate sub-optimal performance. We train one policy on each reward function for a total of 1 million timesteps. We then roll out the policies at each checkpoint to generate 600 demonstrations, that are used to select snippets of length 150 that are ranked using log-likelihoods of the trajectory snippets under the optimal policy. These preferences are fed to the preference learning system.

The experiments follow a very similar outline to the Minigrid experiments outlined in Appendix C.2 above. All frameworks use the same network architecture: A simple two layer Neural Network with 1024 hidden nodes. For the `button-press-v2` env, for example, this policy has 35 input nodes, 1024 hidden nodes, and 4 output nodes, with ReLU activation at the hidden layer. For both MultiCPL and POPL, we pre-train with behavior cloning directly from the demonstrations for 1000 iterations at a batch size of 16 and learning rate of 0.001. We use last layer finetuning for both POPL and MultiCPL. For MultiCPL, we use the CPL objective, and train the last layer using a batch size of 16, learning rate of 0.001, and for 50 iterations each. For POPL, we sample 512 preferences every 25 iterations, and sub sample a batch of size 256 to use for lexicase selections. We use a Gaussian mutation with mean 0 and standard deviation of 0.01 to mutate our policies at each step.

## C.4 LANGUAGE MODEL EXPERIMENTS

In the LLM experiments, we assess the performance of reward learning by examining preference accuracy on the test set. To investigate vulnerabilities to jailbreak, we analyze pairs of responses to jailbreak prompts designed by Wei et al. (2024) to deceive the model into giving a harmful response. We calculate the percentage of prompts where it assigns a higher reward to the jailbroken response ("jailbreak rate"). Additionally, we evaluate the reward function's ability to assess helpfulness on non-harmful prompts, *i.e.*, the reward function predicts higher rewards on the more helpful response. We compare our method to normal RLHF with an LLM-based preference model, Bayesian Reward Extrapolation (B-REx), and distributional preference learning (DPL). DPL methods predict parame-

ters of the distribution of reward values for each response, rather than a single reward value, in order to better account for hidden context in human preferences.

For standard RLHF, we use the pre-trained LLAMA-2-7b (Touvron et al., 2023) preference model by Siththaranjan et al. (2023), which is fine-tuned on the HH-RLHF dataset using LoRA (Hu et al., 2022). We implement B-REx by performing linear reward extrapolation on the last layer of the pre-trained LLAMA-2-7b preference model. Following the B-REx implementation in (Brown et al., 2020a), we run 200,000 steps of MCMC with a step size of 0.05. We use a burn-in of 5000 and a skip every 20 samples to reduce auto-correlation. For POPL, we run lexicase selection for 100 generations with a population size of 1000, and randomly sample 100 reward functions in the last generation.

Because the ranking likelihood is invariant to affine transformations of the rewards, we normalize the rewards by subtracting the median reward calculated on the training set over all the responses. This ensures that the reward values are comparable when calculating the lower quantile of rewards in risk-averse optimization.

## D  BROADER SOCIETAL IMPACTS

The proposed work on Pareto Optimal Preference Learning (POPL) aims to enhance the alignment of AI systems with diverse human values, thereby addressing critical issues of fairness and representation. By focusing on learning from human preferences with hidden context, our method seeks to ensure that AI models do not disproportionately favor or disadvantage specific groups, making them more equitable and just. This has the potential to significantly improve the societal acceptance and trust in AI systems, particularly in sensitive applications such as healthcare, education, and law enforcement, where fairness and inclusivity are critical.

However, there are potential negative societal impacts to consider. The deployment of AI systems that can cater to specific groups might inadvertently reinforce existing biases if the hidden context reflects social prejudices or discriminatory practices. Therefore, it is crucial to incorporate safeguards and robust validation mechanisms to detect and mitigate any biased outcomes. As researchers and developers, we must be vigilant about the sources of our training data and continually audit AI systems for unintended consequences.

Moreover, the computational work required for training these models can have environmental impacts, given the high-energy consumption associated with large-scale AI computations. Researchers should consider optimizing algorithms to be more efficient and exploring the use of renewable energy sources to mitigate this impact.

By considering these factors, we aim to advance AI technologies in a direction that promotes fairness, inclusiveness, and sustainability, ensuring that they serve the broader interests of society.

