# OpenReview forum: "Pareto-Optimal Learning from Preferences with Hidden Context"
_ICLR.cc/2025/Conference — Submitted to ICLR 2025_

### Official Review · Reviewer_isUD · 2024-10-28

**Soundness:** 3
**Presentation:** 3
**Contribution:** 3
**Rating:** 6
**Confidence:** 4

**Summary:**

The paper tries to solve the problem of learning from complex human preference by first formulating it as the problem of preference learning with hidden context, and then relying on the framework and techniques of multi-objective optimization. The Pareto Optimal Preference Learning (POPL) method, based on the lexicase selection algorithm, is proposed to efficiently learn and generate diverse rewards and policies. The extensive and diverse experiments are conducted to illustrate the efficiency and applicability of the method.

**Strengths:**

1. The problems of previous methods are clarified with a simple yet illustrative example.
2. The intuitions and theoretical formulations in Section 5.1 are interesting and may be useful for future research.
3. Experiments are diverse and persuasive, ranging from analytic environments to language model experiments, which show the proposed method’s efficiency and applicability.

**Weaknesses:**

1. It would be better to concisely introduce the theory proposed in Section 4 in the introduction Section.
2. Paragraph headings in Section 2 can make them more clear.
3. In line 147, the comma ahead of ‘However’ should be a full stop.
4. The process of lexicase selection for preference learning in Section 5.2 can be made more clear with figures, formulas, or diagrams.

**Questions:**

1. By formulating the RLHF-HC problem as a multi-objective problem, how will the method deal with (non-)**transitivity** of real-world human preference ? As this could be a problem for real-world human feedback (See ,e.g., section 7.3 of [1]).
2. ‘Theory Foundation’ in the title of Section 4 seems to be missing, or it has been formulated in Section 5 ? A re-organization may be needed.
3. Why the probability of segments in Definition 1 does not include the distributions of the states, such as $\prod_{(s,a)\in\sigma}\pi(a|s)d(s)$ ?
4. In line 258, is the word ‘illicit’ a typo ? It may be ‘elicit’.
5. In the proofs in Appendix B, what does $\pi_z$ refer to ? Is it just $\pi_z^*$ ?

---

> ### Author Response · Authors · 2024-11-19
> **Response to Reviewer isUD**
>
> Dear Reviewer isUD,
>
> Thank you very much for taking the time to read our paper and leave a thoughtful review. Please find answers to your questions below. Thank you for your suggestions to re-organize our manuscript and improve the clarity of our work –- we have implemented all of your suggestions in our manuscript.
>
> > It would be better to concisely introduce the theory proposed in Section 4 in the introduction Section.
>
> Thank you for the suggestion. We have included a brief introduction to the theory proposed, and moved it to the beginning of Section 4.
>
> > Paragraph headings in Section 2 can make them more clear.
>
> We have added section headings for the related work in Section 2. Thank you for the suggestion!
>
> > In line 147, the comma ahead of ‘However’ should be a full stop.
>
> Thank you for catching this. We have fixed it.
>
> > The process of lexicase selection for preference learning in Section 5.2 can be made more clear with figures, formulas, or diagrams.
>
> We have added a new figure to section 5 to outline lexicase selection clearly. The figure outlines the process of lexicase selection to select a single candidate reward function from a set of them, given a randomly ordered set of preferences. We hope that this can help make the section more clear.
>
> # Questions
> > By formulating the RLHF-HC problem as a multi-objective problem, how will the method deal with (non-)transitivity of real-world human preference ? As this could be a problem for real-world human feedback (See ,e.g., section 7.3 of [1]).
>
> The non-transitivity of human preferences is a very important factor that we believe POPL handles well. Let’s examine the case of three preferences that are intransitive. Under any reward function hypothesis, at most two of the three preferences will be satisfied at a time (cannot satisfy all three as they are intransitive). POPL would generate three hypotheses, each satisfying a different two of the three (if possible), as these are the largest non-contradictory subsets. In treating these hypotheses fairly, we avoid over-emphasizing any one of the hypotheses as the Bradley-Terry model would have us do.
>
> > ‘Theory Foundation’ in the title of Section 4 seems to be missing, or it has been formulated in Section 5 ? A re-organization may be needed.
>
> Thanks for the suggestion. We have reorganized the paper as follows:
> - Moved our theoretical grounding from Section 5 to Section 4
> - Moved our discussions of MDPLs to the preliminaries section.
>
> > Why the probability of segments in Definition 1 does not include the distributions of the states, such as $\Pi_{(s, a) \in \sigma}\pi(a|s)d(s)$ ?
>
> Thanks for the question. Definition 1 relies on $(s, a)$ pairs that are sampled from our trajectories $\sigma_i$, meaning that they are implicitly already weighted by the prevalence of the states in the trajectories (and the trajectories are weighted by probability under the demonstration policy)
>
> > In line 258, is the word ‘illicit’ a typo ? It may be ‘elicit’.
>
> Yes, this was a typo. Thanks for pointing it out!
>
> > In the proofs in Appendix B, what does $\pi_z$ refer to? Is it just $\pi_z^*$?
>
> Thanks for pointing this out too! Yes, $\pi_z$ was meant to be $\pi^*_z$ in the proof.

---

> > ### Comment · Reviewer_isUD · 2024-11-24
> > **Thanks for your responses**
> >
> > Thanks for your responses, and I keep my score.

---

### Official Review · Reviewer_7xUX · 2024-10-30

**Soundness:** 3
**Presentation:** 2
**Contribution:** 2
**Rating:** 5
**Confidence:** 3

**Summary:**

POPL enhances alignment by addressing diverse human preferences as multiple objectives, achieving solutions that are Pareto-optimal and fair across groups. It uses lexicase selection to find policies that respect varied preferences across. Empirical results show POPL’s effectiveness in aligning AI to human values safely and equitably across applications like robotics and language models.

**Strengths:**

- This paper provide a novel way to deal with the heterogenity among different sourses
- Various experiments under different settings are conducted to evaluate the performance

**Weaknesses:**

- Noations and concepts are sometimes not well-defined. For instance, $\sigma$ first occur in Section 3 without any definitions. I would suggest the authors to more clearly defined notations.
- The presentation of experiment results is quite unclear. For example, Figure 3(b) is really chaotic and I can hardly tell the information here.
- Lack of comparision with other methods. From my interpretation, many methods are proposed to solve the similar issue, e,g., Nash learning for RLHF and general preference framework. The methods should be more properly evaluated and compared.

**Questions:**

-  What's the formal definition of $\sigma$, is it always a $(s,a)$ pair?
- Could authors provides more clearer highlight of the technical contributions?
- For me, even though the definitions are provided, the concept is still confused. Could authors gives the mathemitical definitions of concepts and theorems in Section 5.1, e.g., optimality?

---

> ### Author Response · Authors · 2024-11-19
> **Response to Reviewer 7xUX**
>
> Dear Reviewer 7xUX,
>
> Thank you very much for reviewing our paper and for your comments and questions. Please find our responses to your specific questions and concerns below.
>
> > Noations and concepts are sometimes not well-defined. For instance, \sigma first occur in Section 3 without any definitions. I would suggest the authors to more clearly defined notations.
>
> At the bottom of that paragraph, we explain “The probability of pairwise preference $(\sigma_i \succ \sigma_j)$ between two trajectories [or segments] given some utility function $f(\sigma)$ can be written as ….“. We have formally defined it further by saying “We operate in a common RLHF setting in which, given a dataset $D = \{ \sigma_1, \cdots, \sigma_m \}$ of trajectory segments and a set $\mathcal{P} = \{(i, j): \sigma_i \succ \sigma_j\}$ of pairwise preferences over these segments,”.
>
> > The presentation of experiment results is quite unclear. For example, Figure 3(b) is really chaotic and I can hardly tell the information here.
>
> We have significantly simplified figure 3 to help ensure accessibility while still preserving the main conclusions.
>
> > Lack of comparision with other methods. From my interpretation, many methods are proposed to solve the similar issue, e,g., Nash learning for RLHF and general preference framework. The methods should be more properly evaluated and compared.
>
> Nash learning builds a pairwise preference model that is conditioned on a pair of inputs and predicts the preference that a person might give. Then, to select an output, their algorithm uses the Nash equilibrium. This is a different problem setting to ours, as we are focusing on learning reward models or policies that map to ground truth reward models of groups of people in our dataset. On the other hand, Nash learning is attempting to choose outputs that are in some sense optimal for all groups. Furthermore, Nash learning is defined in a contextual bandit setting, and is not well defined for environments that are time based such as the Minigrid and Metaworld results we are using in this work.

---

> ### Author Response · Authors · 2024-11-19
> **Continued - Response to Reviewer 7xUX**
>
> Thank you for your questions! Please find answers to them below:
>
> > What's the formal definition of $\sigma$, is it always a $(s,a)$ pair?
>
> Yes, we define $\sigma$ as a segment of a trajectory, or a series of $(s, a)$ pairs that are encountered sequentially during the same episode.
>
> > Could authors provides more clearer highlight of the technical contributions?
>
> Yes, thank you for the suggestion. We will add the following bullet points to the end of the introduction of our paper:
>
>
> - We extend the problem of Reinforcement Learning from Human Feedback with Hidden Context (RLHF-HC) introduced by [1], addressing critical limitations in handling preferences for sequential domains, as opposed to contextual bandits.
> - We derive theoretical results proving that optimal reward functions and policies for hidden context groups are inherently Pareto-Optimal with respect to the given preferences, establishing a rigorous mathematical basis for our approach.
> - We develop ``Pareto-Optimal Preference Learning" (POPL), a framework leveraging lexicase selection to generate a set of Pareto-Optimal reward functions or policies. POPL ensures diverse, group-specific alignment with human preferences, enabling robust personalization and fairness.
> - We demonstrated POPL's superiority over strong baselines across diverse settings, including:
>   -  Minigrid RL: Policy learning in grid-based decision making domains.
>   -   Metaworld:  Balancing safety and speed in 3D robotics manipulation tasks.
>   -   LLM Jailbreaking Detection: Mitigating harmful outputs by aligning preferences for both helpfulness and harmlessness (without labels)
> - We showcased POPL's ability to efficiently scale to high-dimensional tasks, such as those involving LLMs, while maintaining computational efficiency. POPL achieves robust results with pre-trained models, making it broadly applicable across domains requiring fairness, alignment and diversity.
>
> > For me, even though the definitions are provided, the concept is still confused. Could authors gives the mathemitical definitions of concepts and theorems in Section 5.1, e.g., optimality?
>
> Thank you for the question. We define Pareto-optimality in Definition 2 of our paper, where we say that:
>
>  A policy $\pi(a|s): \mathcal{S} \times \mathcal{A} \mapsto [0, 1]$ is Pareto optimal with respect to a set of preferences $\mathcal{P}$ relative to a set of other policies $\Pi = \{\pi_1, \pi_2, \cdots, \pi_n\}$ if and only if there exists a preference $(\sigma_i \succ \sigma_j) \in\mathcal{P}$ that only $\pi$ passes out of all policies in $\Pi$
>
> We also define an optimal policy for a hidden context group in Definition 4:
>
> An optimal policy $\pi^*_g$ for a hidden context group $g$ is an optimal policy in a given environment (MDP/R) using the group's implicit ground truth reward function $r_g$ as the reward function.
>
> The definition of an optimal policy in RL is simply a policy that, for every state $s\in\mathcal{S}$ in the set of all states, has a higher expected return than all other policies [2]. And equivalently, an optimal reward function is simply a reward function that ranks all states equivalently to the ground truth reward function that generated the preferences. In other words, a reward function $r$ is optimal if for every preference $(\sigma_i, \sigma_k) \in \mathcal{P}$, $\sum\limits_{(s, a) \in \sigma_i}r(s, a) > \sum\limits_{(s, a) \in \sigma_j}r(s, a)$. The optimal reward function for a hidden context group is a reward function that is optimal when only considering the preferences coming from members of that hidden context group.
>
> [1] Siththaranjan, Anand, Cassidy Laidlaw, and Dylan Hadfield-Menell. "Distributional preference learning: Understanding and accounting for hidden context in RLHF." arXiv preprint arXiv:2312.08358 (2023).
> [2] Sutton, Richard S., and Andrew G. Barto. "Reinforcement Learning: An Introduction. Second." A Bradford Book (2018).

---

> > ### Comment · Reviewer_7xUX · 2024-11-21
> >
> > I would like to thank the authors for their detailed clarifications and the significant improvements made in the revised version of the paper. These updates have helped me better understand the main contributions, and I have accordingly improved my score. However, I remain unconvinced by the theoretical part, as the assumption of a noiseless setting is quite strong, potentially limiting the practical relevance of the results. To better establish optimality, I suggest conducting a more detailed error analysis on uncertainties.

---

### Official Review · Reviewer_jQRr · 2024-11-04

**Soundness:** 2
**Presentation:** 2
**Contribution:** 2
**Rating:** 5
**Confidence:** 2

**Summary:**

This work concentrates on the issue that human feedback data might involve noises caused by hidden information. It proposes a method named POPL that aims to learn a Pareto-optimal policy. POPL uses lexicase selection and conducts experiments on different tasks.

**Strengths:**

1. This work concentrates on an import problem that human feedback involves hidden information. The Pareto-optimal is indeed one possible solution.

2. Experiments on different tasks are given.

**Weaknesses:**

1. I found the presentation for the method is quite hard for me to understand. Lexicase selection, as key idea of the method, is not introduced clearly. I am not clear about how this selection method is conducted. Also, I think a figure for process might be better for readers to understand.

2. Similarly, I found that many concepts are used without a clear explanation. For example, I am not clear what "hypotheses" refers to as it first shows in Sec. 6.1 and also in Alg. 1. Also, as MDPL is mentioned, this formal calculation for this method is not give. I cannot under stand the illustration of Fig.3.

3. Further, if more intuition about the method is provided, it might be better for readers to understand this method.

4. Since this method concentrates on the RLHF alignment process, I am curious about the comparison between POPL and more RLHF methods like DPO. Also, the detailed information for the standard RLHF is not given.

**Questions:**

See the weakness part above.

---

> ### Author Response · Authors · 2024-11-19
> **Response to Reviewer jQRr**
>
> Dear Reviewer jQRr,
>
> Thank you very much for taking the time to read our paper. We have responded to your questions and concerns below. Please don't hesitate to ask for clarification further if needed!
>
> > I found the presentation for the method is quite hard for me to understand. Lexicase selection, as key idea of the method, is not introduced clearly. I am not clear about how this selection method is conducted. Also, I think a figure for process might be better for readers to understand.
>
> We have added a figure to our lexicase selection section specifically to outline the lexicase selection process in more depth. In this diagram, we walk through a single selection step, starting with a random preference ordering, and progressively filtering down a population of reward functions based on this random preference ordering. We hope that this will give a good visual of the process of using lexicase selection to select pareto-optimal candidates.
>
> > Similarly, I found that many concepts are used without a clear explanation. For example, I am not clear what "hypotheses" refers to as it first shows in Sec. 6.1 and also in Alg. 1. Also, as MDPL is mentioned, this formal calculation for this method is not give. I cannot under stand the illustration of Fig.3.
>
> Thank you for your feedback. We use the the term “hypothesis” in the standard way that it is commonly used across the ML literature to refer to potential models that explain the observed data. However, we have added a brief explanation in section 6.1 to improve accessibility.
>
> In response to your question on MDPLs, MDPL is not a single method, but a class of methods that rely on marginalization over hidden context in a single model. Mathematically, we introduce them as “rel[ying] on a single model of utility $u(s | z)$ to output a distribution of utility assignments $u(s)$ for each state $s\in \mathcal{S}$, marginalizing over the hidden context variable $z$.” This is sufficient detail that we believed necessary for understanding this class of algorithms for the purposes in our work. For more detail on them, please look at [1]. We have improved the quality of Figure 2 to provide more intuition on MDPLs as well.
>
> Thanks for your comments on Fig.3. We have significantly simplified this figure to avoid confusion and focus on the main takeaways from that experiment.
>
> > Further, if more intuition about the method is provided, it might be better for readers to understand this method.
>
> We have improved our explanations in text to help a reader build intuition regarding the fittingness of our approach.
>
> > Since this method concentrates on the RLHF alignment process, I am curious about the comparison between POPL and more RLHF methods like DPO. Also, the detailed information for the standard RLHF is not given.
>
> In our robotics experiments, we use contrastive preference learning (CPL), which is a more general framework than DPO (specifically, DPO is a specific instance of CPL in the contextual bandit setting). For our LLM experiments, we compare against standard RLHF, the established baseline in this domain. DPO simply consolidates the RLHF pipeline into a single step. Since our focus is on handling hidden context, DPO’s approach does not offer any advantage in this area, making standard RLHF a more relevant baseline. We don’t believe that DPO would add any scientific understanding, as the algorithmic differences between DPO and RLHF don’t interact in an interesting way with the scientific question we are answering.
>
> [1] Siththaranjan, Anand, Cassidy Laidlaw, and Dylan Hadfield-Menell. "Distributional preference learning: Understanding and accounting for hidden context in RLHF." arXiv preprint arXiv:2312.08358 (2023).

---

### Official Review · Reviewer_G5YS · 2024-11-05

**Soundness:** 2
**Presentation:** 1
**Contribution:** 2
**Rating:** 5
**Confidence:** 3

**Summary:**

This paper introduces a reinforcement learning from human feedback with hidden context (RLHF-HC) framework called Pareto optimal preference learning (POPL). While marginalized distributional preference learning (MDPL) marginalizes $u(s\vert z)$ over the hidden context variable $z$ to produce a distribution $u(s)$ for each state $s\in\mathcal{S}$, POPL preserves the full conditional distributions. By leveraging these conditional probabilities, POPL can effectively capture Pareto optimal reward functions.

**Strengths:**

POPL can capture the Pareto optimal reward functions.

**Weaknesses:**

- The validity of the proposed algorithm is not theoretically explained.
- The scalability of the proposed algorithm remains uncertain.
- There is no way to determine the dimensionality of the underlying reward functions or the number of Pareto optimal policies.

**Questions:**

I am unsure if I fully understand the paper. If I have any misunderstandings, please let me know. I would be happy to receive clarification.

1. As I understand it, there is no learning in Algorithm 1, which represents one step of POPL. In my understanding, it solely relies on the random initialization of hypotheses and the selection of good hypotheses. With a large number of trials, it might eventually find good policy hypotheses, but I wonder if this method is truly practical.
2. In general, there are many policies that passes exactly the same preference subsets. However, Algorithm 1 only checks whether the candidates pass at least on preference or not. Then, how to determine a Pareto-optimal policy among these policies?
3. In Algorithm 1, if we use POPL to learn a set of reward functions, how can we specify the dimensionality of these reward functions and learn them from the given demonstrations?
4. Compared to the previous works, such as preference-driven MORL [1] or multi-objective alignment in LLM [2], what is the main difference and advantage of POPL?
5. I cannot understand the left side of Figure 2 (titled “MDPLs”). In this case, what does z indicate?

### References

[1] Basaklar, Toygun, Suat Gumussoy, and Umit Y. Ogras. "Pd-morl: Preference-driven multi-objective reinforcement learning algorithm." ICLR 2023.

[2] Yang, Rui, et al. "Rewards-in-context: Multi-objective alignment of foundation models with dynamic preference adjustment." arXiv preprint arXiv:2402.10207 (2024).

---

> ### Author Response · Authors · 2024-11-19
> **Response to Reviewer G5YS**
>
> Dear Reviewer G5YS,
>
> Thank you for taking the time to review our paper.
>
> Weaknesses:
> > The validity of the proposed algorithm is not theoretically explained.
>
> Lexicase selection as an algorithm is guaranteed to select candidates that are Pareto-optimal with respect to the current population, with a sufficiently diverse population, these individuals will be globally Pareto-Optimal. We link to proofs of that in related work [1]. In their conclusions, they state that “individuals selected by lexicase selection occupy the boundaries or near boundaries of the Pareto front in the high-dimensional space spanned by the population errors. “. Formally, they prove their Theorem 3.4: "If individuals from a population N are selected by lexicase selection, those individuals are Pareto set boundaries of N with respect to T."
>
> We also prove in the theory sections of our paper that optimal policies and reward functions for distinct hidden context groups are Pareto-optimal. This, in our view, is theoretical grounding for why our work is valid.
>
> [1] William La Cava, Thomas Helmuth, Lee Spector, Jason H. Moore; A Probabilistic and Multi-Objective Analysis of Lexicase Selection and $\epsilon$-Lexicase Selection. Evol Comput 2019; 27 (3): 377–402. doi: https://doi.org/10.1162/evco_a_00224
>
> > The scalability of the proposed algorithm remains uncertain.
>
> We scale our method to high-dimensional domains, including complex robotics tasks and applications with LLMs, demonstrating its scalability. We believe this covers a range of challenging scenarios, but are open to suggestions if there are other benchmarks you feel would strengthen this point further.
>
> > There is no way to determine the dimensionality of the underlying reward functions or the number of Pareto optimal policies.
>
> Picking the dimensionality of underlying reward functions is a problem with preference learning in general, and we do not expect our method to act differently to other RLHF methods. In our experiments, we used the same setup as that used by our baselines (B-Rex, CPL, DPL, etc) with promising performance improvements over them. For picking the number of Pareto-optimal policies, POPL automatically finds as many Pareto Optimal policies as possible (given the fixed population size). The size of the population is relatively unimportant, so long as it is larger than the number of hidden context groups. In our experiments, we have used population sizes of 50, 100, 500 and 1000 with no noticeable effects on performance or experiment time.

---

> ### Author Response · Authors · 2024-11-19
> **Continued - Response to Reviewer G5YS**
>
> # Questions
>
> > As I understand it, there is no learning in Algorithm 1, which represents one step of POPL. In my understanding, it solely relies on the random initialization of hypotheses and the selection of good hypotheses. With a large number of trials, it might eventually find good policy hypotheses, but I wonder if this method is truly practical.
>
> We would like to clarify that Algorithm 1 indeed incorporates learning, in which candidate hypotheses are selected and refined based on observed data.
>
> In Algorithm 1, each iteration generates a set of hypotheses and selectively retains only those that perform well on subsets of the preference data, thereby gradually refining the hypothesis space over time. This filtering process guides the search toward increasingly diverse and effective policies without needing a conventional gradient-based learning method. As in Bayesian filtering, where a belief distribution is updated based on observed evidence, our approach continually narrows the hypothesis space based on selection criteria, effectively ‘learning’ which policies hold promise given the current data. We have added a new figure to outline a single step of lexicase selection to help illustrate this point further.
>
> > In general, there are many policies that passes exactly the same preference subsets. However, Algorithm 1 only checks whether the candidates pass at least on preference or not. Then, how to determine a Pareto-optimal policy among these policies?
>
> Thanks for your question. We would like to clarify that Algorithm 1 does not only check whether the candidates pass at least one preference or not. Instead, Algorithm 1 checks whether the candidates pass one preference at a time, in a random order, until there is only one left. In practice, this means that every preference exerts some selection pressure at different points in the selection procedure, which results in provably Pareto-optimal policies..
>
> > Compared to the previous works, such as preference-driven MORL [1] or multi-objective alignment in LLM [2], what is the main difference and advantage of POPL?
>
>  Preference-driven MORL is not operating in a domain with pairwise human preferences, it instead considers preferences (weightings) over objectives. In that work, the ground truth objectives are determined in advance and are fed as input to the MORL system. This is a fundamentally different problem setting to that used in our work. Similarly, "Rewards-in-context: Multi-objective alignment of foundation models with dynamic preference adjustment” focuses on aligning to multiple objectives that are already given. POPL’s unique advantage is that it is able to infer reward functions and policies that are optimal for groups of preference givers, without having labels for group membership nor handwritten objectives to optimize.
>
> > I cannot understand the left side of Figure 2 (titled “MDPLs”). In this case, what does z indicate?
>
> In our Figure 2, $z$ was meant to denote the hidden context of an individual. We have adapted Figure 2 to be more clear regarding this.

---

> > ### Comment · Reviewer_G5YS · 2024-11-27
> >
> > Thank you for your detailed response. This rebuttal helped me gain a better understanding of this work, and I have raised my score.
> >
> > However, I still have unresolved questions, including concerns about the theoretical justification and practical applicability.
> >
> > Therefore, I cannot raise my score above 5.

---

### Author Response · Authors · 2024-11-19
**Official Comment by Authors**

Dear Reviewers,

Thank you all for taking the time to carefully review our paper and provide thoughtful feedback. we truly appreciate your suggestions and have worked to address all concerns raised to improve the clarity, structure and depth of our manuscript.

We are encouraged by your recognition of the importance of our work and its potential impact. For example, various reviewers mention that we focus on "an import[ant] problem" (Reviewer jQRr) and that POPL is a novel solution for this problem (Reviewer 7xUX). We are also happy to hear that you believe the theory we performed to be interesting and helping to direct future research (Reviewer isUD). Furthermore, we are thrilled by the recognition that our "experiments are diverse and persuasive, ranging from analytic environments to language model experiments, which show the proposed method’s efficiency and applicability.".

In response to the suggestions brought forward by various reviewers, we have incorporated the following changes to our paper:
- Restructuring of the presentation of our work to help improve flow and explanation
- We updated the figure outlining the issues with the use of MDPLs to help improve its accessibility.
- We simplified figure 3 (now figure 4) to concentrate on the main take-aways.
- We added aggregative results for the Metaworld experiments, to help show how our method scales and is robust to stochasticity.
- Expanded the introduction with concise highlights of our theoretical contributions and technical novelty.

We believe that these changes make our paper stronger and address the concerns raised in your reviews. Thank you again for your constructive feedback, and we hope you find the revised version compelling.

Sincerely,
Authors

---

### Comment · Area_Chair_XM1X · 2024-11-25

Dear Reviewers,


This is a friendly reminder that the discussion will end on Nov. 26th (anywhere on Earth). If you have not already, please take a close look at all reviews and author responses, and comment on whether your original rating stands.


Thanks,

AC

---

### Meta-Review · Area_Chair_XM1X · 2024-12-20

**Metareview:**

This paper proposes Pareto Optimal Preference Learning (POPL) to enable pluralistic alignment by framing discrepant group preferences as objectives with potential trade-offs, aiming for policies that are Pareto-optimal on the preference dataset.

The proposed Pareto-optimal solution is a promising direction for improving RLHF alignment and The concept of Pareto optimality is well-integrated into the framework. It also includes a wide range of experiments across different tasks.

The main concern is the validity of the proposed algorithm is not sufficiently supported by theoretical explanations. There is a need for more formal justification to back up the claims made about the algorithm's effectiveness. Moreover, the scalability of the proposed algorithm is not clearly addressed. It remains uncertain how well the method will perform as the complexity of the task or the number of reward functions increases.

**Additional Comments On Reviewer Discussion:**

Two reviewers are still concerned about theoretical justification and practical applicability after rebuttal.

---

### Decision · Program_Chairs · 2025-01-22

Reject